# KVLinC: KV Cache Quantization with Hadamard Rotation and Linear Correction

## Abstract

Quantizing the key-value (KV) cache is a promising strategy for improving the inference efficiency of large language models (LLMs). However, aggressive quantization to very low precision (e.g., 2 bits) introduces significant errors in the stored key and value tensors, which propagate through the dot-product attention mechanism and ultimately degrade generation quality. To address this, we propose *KVLinC*, a framework to mitigate attention errors introduced by KV cache quantization in the extreme low-precision regime. KVLinC combines a Hadamard rotation, which reduces quantization error in values, with lightweight linear correction adapters that explicitly compensate for errors introduced by quantized keys. Across extensive evaluations on the LLaMA, Qwen2.5, and Qwen3 model families, KVLinC consistently matches or surpasses strong baselines while achieving higher KV-cache compression. Furthermore, we implement a custom attention kernel that results in upto $2.55\times$ faster inference compared to Flash Attention baseline, enabling efficient long-context LLM inference.

## 1 Introduction

Large Language Models (LLMs) (Meta, 2024a;b; Yang et al., 2024a; 2025) have achieved strong performance across diverse NLP tasks, but their deployment remains costly due to heavy memory and compute demands during inference. A major bottleneck is the key-value (KV) cache, which stores past activations in every transformer layer to enable autoregressive decoding. Unlike model parameters, which are fixed in size, the KV cache grows linearly with sequence length and batch size, quickly dominating GPU memory and bandwidth. For example, in Llama-3-8B (Meta, 2024b) with a sequence length of 8k and a batch size of 16, the KV cache alone consumes 16 GB, which is comparable to the parameter footprint. As applications push toward longer contexts or larger batch sizes, the KV cache quickly dominates memory and bandwidth requirements, limiting throughput and inflating serving costs. Thus, reducing KV cache size while preserving accuracy is critical for scaling LLMs to long-context and high-throughput settings.

Quantization of KV cache is a promising direction to reduce inference memory cost by representing the key value tensors in lower precision formats (Hooper et al., 2024; Liu et al., 2024b; Ashkboos et al., 2024). Recent work KIVI (Liu et al., 2024b) has demonstrated the feasibility of compressing the KV cache to as few as 2-bits per entry. However, quantizing the KV cache to low precision introduces quantization errors in the stored key and value tensors which propagate into the dot-product attention mechanism and ultimately impair language generation ability. As sequence length of a task increases, quantization errors accumulate across the stored key and value tokens, leading to compounding distortions in attention distributions. Since each decoding step reuses the corrupted representations, performance degradation becomes more severe with increasing sequence length Kang et al. (2024).

QuaRot (Ashkboos et al., 2024) demonstrated that applying a rotation prior to quantization can substantially reduce quantization error compared to directly quantizing the raw tensor. Specifically, QuaRot leverages Hadamard rotations to rotate the key and value tensors into a representation more suitable for low-precision storage. While this approach has shown effectiveness at moderate precision levels, such as a 4-bit KV cache, its applicability under more aggressive quantization settings remains unexplored. In contrast, another line of work focuses on compensating for quantization error by preserving selected components of the KV cache in higher precision. For example, ResQ

(Saxena et al., 2024b) retains critical channels in high precision, while Gear (Kang et al., 2024) maintains a low-rank reconstruction of the quantization error. However, in both cases, the memory cost of storing high-precision components grows proportionally with the KV cache. At long context lengths, this overhead becomes non-negligible, limiting the overall compression benefits of KV cache quantization.

To address this, we propose KVLinC, a framework explicitly designed to mitigate attention errors introduced by KV cache quantization in the extreme low-precision regime. KVLinC combines complementary strategies for keys and values that enable robust compression of the KV cache to 2-bit while maintaining strong performance across both short and long context tasks. First, we revisit rotation-based quantization methods and analyze their robustness at 2-bit precision. We explore different quantization axes — specifically, applying quantization along the channel axis or the token axis when combined with Hadamard rotated keys and values. Our experiments reveal that optimal performance is achieved by quantizing raw keys along the channel axis, while rotated values perform best when quantized along the token axis.

Second, to further mitigate the impact of quantization error, we introduce linear correction adapters, trainable modules that explicitly learn to track and compensate for distortions in the attention distribution caused by quantized keys. These adapters incur only a constant memory overhead that does not grow with sequence length. Moreover, their computational cost is linear with sequence length, in contrast to quadratic complexity of self-attention, making them both efficient and practical for long-context inference. Our design is motivated by linear attention methods (Zhang et al., 2024; Lan et al., 2025), which discard most tokens and train adapters to recover the resulting error. While effective for short contexts, such methods replace softmax with a lossy linear approximation, leading to distortions that cannot be fully corrected. In contrast, our approach retains the full token history and corrects only quantization-induced errors in keys which makes it an easier learning problem. This allows KVLinC to achieve effective compression while preserving the fidelity of softmax attention, naturally scaling to long contexts. In summary, our contributions are as follows:

- We analyze the various design choices related to Hadamard rotation based KV cache quantization and observe that quantizing keys along the channel axis and quantizing Hadamard rotated values along the token axis is optimal.

- We introduce linear correction adapters which are trained to correct attention error introduced by quantized KV cache.

- We evaluate KVLinC on various short and long context benchmarks for base and instruct models and show that KVLinC either matches or achieves superior performance with higher KV cache compression.

- We develop a Triton (Tillet et al., 2019) based attention decoding kernel which along with off-the-shelf quantization kernel achieves up to $2.55\times$ faster inference and up to $3.5\times$ larger batch size with KVLinC.

## 2 BACKGROUND

**Quantization.** In asymmetric integer quantization, the full-precision tensor $\boldsymbol{X}_r$ is first mapped to an integer representation $\boldsymbol{X}_I$ as :

$$\boldsymbol{X}_I = \left\lfloor \frac{\boldsymbol{X}_r - z}{s} \right\rceil, \qquad s = \frac{\max(\boldsymbol{X}_r) - \min(\boldsymbol{X}_r)}{2^N - 1}, \quad z = \min(\boldsymbol{X}_r), \quad (1)$$

and dequantized as : $Q(\boldsymbol{X}) = \boldsymbol{X}_q = s\boldsymbol{X}_I + z$, where $\boldsymbol{X}_I \in [0, 2^N - 1]$ are $N$-bit integers, $s$ is the scaling factor of quantization and $z$ is the zero-point. Quantization can be performed per tensor where $s$ and $z$ are scalars obtained for the entire tensor or, group-wise where $G$ consecutive entries share a scale factor and zero-point. Group-wise quantization reduces quantization error but requires storing multiple scale factors and zero-points. For $\boldsymbol{X} \in \mathbb{R}^{n \times d}$, channel-wise quantization ($Q_C(\boldsymbol{X})$) groups entries by column $j$ and token-wise ($Q_T(\boldsymbol{X})$) by row $i$ as shown in Figure 1. For assymetic integer quanti-

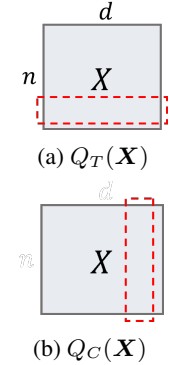

(a) $Q_T(\boldsymbol{X})$

(b) $Q_C(\boldsymbol{X})$

Figure 1: Token-wise and channel-wise quantization grouping.

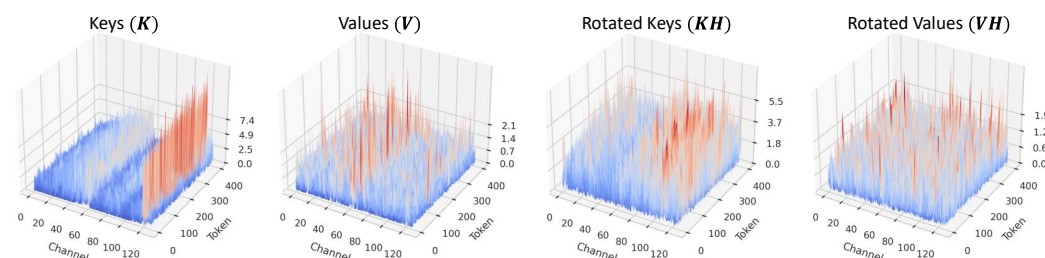

Figure 2: Distribution of key and values with and without Hadamard rotation for Qwen-2.5-3B layer 16 head 0.

zation, the quantization error is given by (Peters et al., 2023) :

$$\mathbb{E}\big[(Q(\boldsymbol{X}) - \boldsymbol{X})^2\big] = \frac{s^2}{12} \tag{2}$$

**Multi Head Attention.** A typical LLM consists of $L$ decoder layers with each layer containing a multi head attention and a feed forward network module. The multi head attention module computes attention per head in parallel with each attention head computing $\boldsymbol{Y} \in \mathbb{R}^{N \times d}$ from inputs $\boldsymbol{X} \in \mathbb{R}^{N \times d}$ (where $N$ is sequence length and $d$ is head dimension) with query, key and value weights $\boldsymbol{W}_q, \boldsymbol{W}_k, \boldsymbol{W}_v \in \mathbb{R}^{d \times d}$. First, we compute $\boldsymbol{Q} = \boldsymbol{X}\boldsymbol{W}_q, \boldsymbol{K} = \boldsymbol{X}\boldsymbol{W}_k, \boldsymbol{V} = \boldsymbol{X}\boldsymbol{W}_v$, before getting attention weights $\boldsymbol{A}$ and attention outputs $\boldsymbol{Y}$ as

$$\boldsymbol{A}_{n,i} = \frac{\exp\left(\boldsymbol{Q}_n \boldsymbol{K}_i^\top / \sqrt{d}\right)}{\sum_{i=1}^n \exp\left(\boldsymbol{Q}_n \boldsymbol{K}_i^\top / \sqrt{d}\right)}, \quad \boldsymbol{Y}_n = \sum_{i=1}^n \boldsymbol{A}_{n,i} \boldsymbol{V}_i, \quad \text{for } n \text{ in } [1, \dots, N] \tag{3}$$

The final output is obtained by concatenating $\boldsymbol{Y}$ across $h$ heads and using output projection matrix $\boldsymbol{W}_o \in \mathbb{R}^{hd \times hd}$ to compute $\boldsymbol{O} = [\boldsymbol{Y}^1, \dots \boldsymbol{Y}^h]\boldsymbol{W}_o$.

**LLM Inference.** LLM inference proceeds in two phases: prefill and decoding. In the prefill phase, per-head token embeddings have shape $\mathbb{R}^{n_p \times d}$, where $n_p$ is the prompt length. The attention computes queries, keys, and values for the prompt and caches the keys and values for subsequent steps. During decoding, the model generates $n_g$ tokens autoregressively, one at a time. At each step $t$ with $n_p < t \le n_p + n_g$, the model forms the new token embedding $\boldsymbol{X}_t$, computes $(\boldsymbol{Q}_t, \boldsymbol{K}_t, \boldsymbol{V}_t) \in \mathbb{R}^{1 \times d}$, and appends $\boldsymbol{K}_t$ and $\boldsymbol{V}_t$ to the cache, yielding $[\boldsymbol{K}_0, \dots, \boldsymbol{K}_t]$ and $[\boldsymbol{V}_0, \dots, \boldsymbol{V}_t]$. Multi-head attention then uses $\boldsymbol{Q}_t$ to attend over the cached keys/values. With KV cache quantization, the cache stores quantized keys and values together with their scale and zero-point parameters, and these are dequantized before the attention computation.

## 3 METHODOLOGY

In this section, we introduce KVLinC, a framework for mitigating attention errors due to low-precision KV cache quantization. KVLinC integrates two complementary strategies: (i) Hadamard rotation to reduce quantization error and (ii) lightweight linear correction adapters to compensate attention distortions. We analyze axis and rotation choices for quantization, describe the design and efficiency of correction adapters, and present a custom attention kernel for accelerated decoding. These components together enable robust long-context inference at low precision with minimal overhead.

### 3.1 HADAMARD ROTATION AND KV CACHE QUANTIZATION

Key and value tensors in the KV cache follow different statistics, motivating distinct quantization schemes. As shown in Figure 2, keys contain channel-wise outliers with a few disproportionately large magnitudes, whereas values do not. KIVI Liu et al. (2024b) addresses this by quantizing keys channel-wise and values token-wise, yielding $\boldsymbol{K}_q = Q_C(\boldsymbol{K}), ; \boldsymbol{V}_q = Q_T(\boldsymbol{V})$. This aligns the

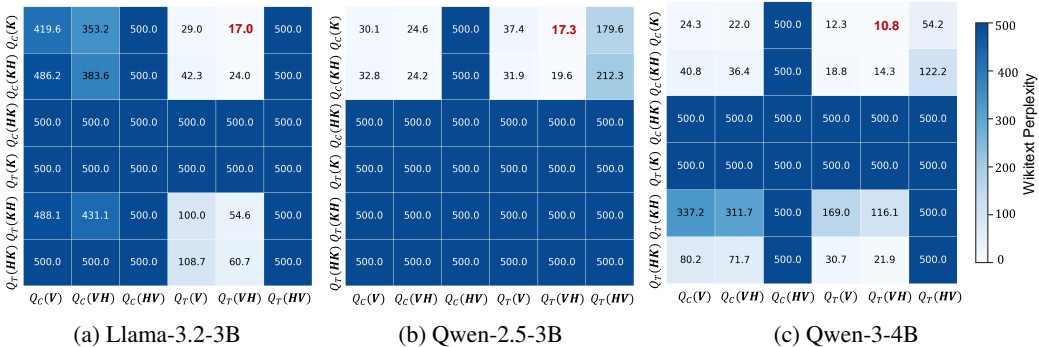

(a) Llama-3.2-3B        (b) Qwen-2.5-3B        (c) Qwen-3-4B

Figure 3: Wikitext perplexity under different 2-bit quantization configuration for key and values. Perplexity values are clipped to $500$. Quantizing raw keys channel-wise and quantizing Hadamard rotated values token-wise achieves best performance (shown in red).

dynamic range per column, localizing key quantization error to individual channels and matching the observed outlier structure. In contrast, QuaRot (Ashkboos et al., 2024) employs a Hadamard rotation to suppress outliers and quantizes both keys and values token-wise. Denoting the Hadamard matrix by $H$, the quantization configuration is $K_q = Q_T(KH), V_q = Q_T(VH)$. As shown in Figure 2, the Hadamard transform equalizes key and value distributions, eliminating outliers, though its effectiveness under extreme low-precision remains untested. During dequantization, the quantized tensors $K_q$ and $V_q$ must be multiplied by $H^\top$, the inverse of the orthogonal Hadamard matrix, introducing additional computational overhead. While the overhead associated with values can be eliminated by merging the rotation into the projection weight matrices, keys still require online Hadamard transforms at inference time. QuaRot applies a Hadamard transform by post-multiplying keys and values before quantization ($KH, VH$); we also consider pre-multiplication ($HK, HV$). This yields a two-dimensional design space: quantization axis (channel- vs. token-wise) × Hadamard placement (pre vs. post). We ablate all combinations, quantizing $K$ and $V$ to 2-bit with group size 128, and evaluate Wikitext perplexity across three model families (Fig. 3). We make the following observations:

**Observation 1.** Pre-multiplying keys and values with a Hadamard matrix almost always yields worse performance compared to post-multiplication. A likely explanation is that pre-multiplication mixes tokens prior to quantization, which amplifies quantization noise and injects errors into the attention logits. In contrast, post-multiplication only rotates channels within each token, thereby preserving relative token alignment and resulting in significantly more stable performance. We also provide layerwise attention error in Figure 8.

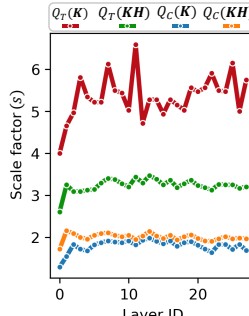

**Observation 2.** At the low-precision regime under consideration, KIVI's quantization configuration consistently outperforms QuaRot's. QuaRot exhibits extremely high perplexity, suggesting that token-wise quantization of keys therefore still incurs large errors. To analyze quantization error (eq. 2), we analyze the scaling factor for different quantization configuration of keys in Figure 4. It shows that although, Hadamard rotation of keys reduces scaling factor and hence the quantization error with token-wise quantization, it still is much higher than channel-wise quantization of keys.

Figure 4: Layer-wise scaling factor for different quantization configuration of keys.

**Observation 3.** Quantizing raw keys channel-wise together with Hadamard rotated values token-wise ($K_q = Q_C(K), V_q = Q_T(VH)$) emerges as the optimal configuration across all model families. This even outperforms the $K_q = Q_C(KH), V_q = Q_T(VH)$ quantization scheme. This is because the application of Hadamard rotation to keys redistributes each outlier dimension, thereby increasing the scaling factor of quantization leading to higher error (Figure 4). We therefore adopt this configuration for KVLinC quantization. Importantly, this scheme is not only optimal in terms of

accuracy but also practical, as it requires no additional computational overhead for quantization or dequantization.

## 3.2 LINEAR CORRECTION ADAPTERS

To further mitigate the errors introduced in the attention operation by quantized keys, we propose correction adapters which are lightweight, trainable modules that explicitly learn to compensate for distortions in the attention distribution. Let the quantization error in keys be denoted by $\boldsymbol{K}^e = \boldsymbol{K} - \boldsymbol{K}^q$. We augment the standard attention formulation with additive correction terms in both the numerator and denominator:

$$\hat{\boldsymbol{Y}}_n = \frac{\sum_{i=1}^n \exp\left(\boldsymbol{Q}_n \boldsymbol{K}_i^{q\top}/\sqrt{d}\right) \boldsymbol{V}_i^q + \sum_{i=1}^n f(\boldsymbol{Q}_n, \boldsymbol{K}_i^e)\boldsymbol{V}_i^q}{\sum_{i=1}^n \exp\left(\boldsymbol{Q}_n \boldsymbol{K}_i^{q\top}/\sqrt{d}\right) + \sum_{i=1}^n f(\boldsymbol{Q}_n, \boldsymbol{K}_i^e)}. \tag{4}$$

Given a query, these correction terms add residual attention weights corresponding to the error induced by quantization. By reparameterizing the correction term additively, we obtain a lightweight approximation that captures the dominant error while remaining computationally efficient. Let the correction adapters $\phi_q, \phi_k : \mathbb{R}^d \mapsto \mathbb{R}^D$ be the trainable feature maps. We define the correction term as the dot product of query and key error feature maps: $f(\boldsymbol{Q}_n, \boldsymbol{K}_i^e) = \phi_q(\boldsymbol{Q}_n)\phi_k(\boldsymbol{K}_i^e)^\top$. This allows the numerator of the correction term to be written as $\phi(\boldsymbol{Q}_n)\sum_{i=1}^n \phi_k(\boldsymbol{K}_i^e)^\top \boldsymbol{V}_i^q$, and the denominator as $\phi_q(\boldsymbol{Q}_n)\sum_{i=1}^n \phi_k(\boldsymbol{K}_i^e)^\top$. With $\boldsymbol{S}_0 = \boldsymbol{0}$ and $\boldsymbol{P}_0 = \boldsymbol{0}$, we compute attention as,

$$\hat{\boldsymbol{Y}}_n = \frac{\sum_{i=1}^n \exp\left(\boldsymbol{Q}_n (\boldsymbol{K}_i^q)^\top/\sqrt{d}\right) \boldsymbol{V}_i^q + \phi_q(\boldsymbol{Q}_n)\boldsymbol{S}_n}{\sum_{i=1}^n \exp\left(\boldsymbol{Q}_n (\boldsymbol{K}_i^q)^\top/\sqrt{d}\right) + \phi_q(\boldsymbol{Q}_n)\boldsymbol{P}_n}, \tag{5}$$

for $\boldsymbol{S}_n = \boldsymbol{S}_{n-1} + \phi_k(\boldsymbol{K}_n^e)^\top \boldsymbol{V}_n$ and $\boldsymbol{P}_n = \boldsymbol{P}_{n-1} + \phi_n(\boldsymbol{K}_n^e)$. This recurrent formulation transforms the quadratic accumulation of correction terms into linear-time updates, allowing error compensation to scale efficiently with sequence length. The cost of error correction is $\mathcal{O}(ndD)$ in time and memory during prefill, and only $\mathcal{O}(dD)$ per step during decoding. At decoding time, the cache stores the quantized keys and values along with the correction states $\boldsymbol{S}_n \in \mathbb{R}^{d \times D}$ and $\boldsymbol{P}_n \in \mathbb{R}^D$. The additional memory cost is constant with respect to sequence length, making the correction adapters highly efficient. Following LolCats Zhang et al. (2024), we choose the feature maps $\phi$ as

$$\phi(\boldsymbol{X}) = [\mathrm{softmax}(\boldsymbol{X}\boldsymbol{W}_1), \mathrm{softmax}(\boldsymbol{X}\boldsymbol{W}_2)] \in \mathbb{R}^D \tag{6}$$

with learnable weights $\boldsymbol{W}_1, \boldsymbol{W}_2 \in \mathbb{R}^{d \times D/2}$. The trainable feature maps add less than $1\%$ parameter overhead. The weights are trained such that the full-precision attention weights $\boldsymbol{A}_{n,i}$ (eq. 3) match the corrected quantized attention weights $\hat{\boldsymbol{A}}_{n,i}$:

$$\hat{\boldsymbol{A}}_{n,i} = \frac{\exp\left(\boldsymbol{Q}_n \boldsymbol{K}_i^{q\top}/\sqrt{d}\right) + \phi_q(\boldsymbol{Q}_n)\phi_k(\boldsymbol{K}_i^e)^\top}{\sum_{i=1}^n \exp\left(\boldsymbol{Q}_n \boldsymbol{K}_i^{q\top}/\sqrt{d}\right) + \phi_q(\boldsymbol{Q}_n)\phi_k(\boldsymbol{K}_i^e)^\top}$$

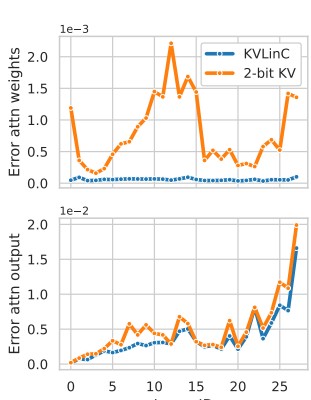

Figure 5: Layer-wise attention error from KV-cache quantization: (top) MSE between full-precision and quantized attention weights; (bottom) MSE between outputs. KVLinC (blue) consistently reduces error versus 2-bit KV (orange).

Using a calibration dataset, we optimize the feature map parameters to reduce the cross-entropy loss between $\boldsymbol{A}_{n,i}$ and $\hat{\boldsymbol{A}}_{n,i}$. As shown in Figure 5, after training, the error between quantized attention and full precision attention is minimized. Thus, correction adapters enable quantized attention to closely match full-precision distributions.

## 3.3 SYSTEM LEVEL IMPLEMENTATION

Improving end-to-end performance with KV-cache quantization requires custom kernels to (1) quantize the cache and (2) run attention directly on quantized operands. We adopt the quantization kernel

---

**Algorithm 1** KVLinC forward pass (single decode step)

---

**Require:** $Q \in \mathbb{R}^{1 \times d}, K_I \in \mathbb{R}^{N/16 \times d}, Z_k, S_k \in \mathbb{R}^{N/G \times d}, V_I \in \mathbb{R}^{N \times d/16}, Z_v, S_v \in \mathbb{R}^{N \times d/G}$,
$C_n, C_d \in \mathbb{R}^{1 \times d}, G$ (quantization group size).
**Ensure:** Output $Y \in \mathbb{R}^{1 \times d}$.
1: Divide $K_I, Z_k, S_k, V_I, Z_v, S_v$ in $T = \lceil N/G \rceil$ blocks: $K_I^1, \ldots, K_I^T$ of size $\frac{G}{16} \times d$ each, $S_k^1, Z_k^1 \ldots, Z_k^T, S_k^T$ of size $1 \times d$ each, $V_I^1 \ldots V_I^T$ of size $G \times \frac{d}{16}$ each and $S_v^1, Z_v^1 \ldots S_v^T, Z_v^T$ of size $G \times \frac{d}{G}$ each.
2: Create empty softmax state $Y_s \in \mathbb{R}^{T \times d}, l, m \in \mathbb{R}^T$
3: Load $Q$ from HBM to SRAM.
4: **parallel For** $j = 1$ to $T$                ▷ Parallelized across KV blocks
5:      Load $K_I^j, S_k^j, Z_k^j, V_I^j, S_v^j, Z_v^j$ from HBM to SRAM.
6:      On chip, dequantize keys: $K_q^{j\top} = \text{unpack}(K_I^{j\top}) \odot S_k^j + Z_k^j$.
7:      On chip, compute $S^j = s \cdot Q K_q^\top$.
8:      On chip, compute $m^j = \text{rowmax}(S^j)$,     $E^j = \exp(S^j - m)$,    $l^j = \text{rowsum}(E^j)$.
9:      On chip, dequantize values : $V_q^j = \text{unpack}(V_I^j) \odot S_v^j + Z_v^j$.
10:     On chip, compute : $Y_s^j = E^j V_q^j \in \mathbb{R}^{1 \times d}$.
11:     Write $Y_s^j, m^j$ and $l^j$ to HBM.
12: **end parallel For**
13: $w = \exp(m - \max(m))$
14: $n = \text{rowsum}(Y_s \cdot w) + C_n$,     $d = (l \cdot w) + C_d$       ▷ attention numerator and denominator
15: $Y = n/d$

---

from KIVI (Liu et al., 2024b), which quantizes the KV cache to 2 bits and bit-packs 16 elements into a single 32-bit word. To accelerate decoding, we implement a custom attention kernel in Triton (Tillet et al., 2019). In the spirit of FlashAttention (Dao, 2023), the kernel streams blocks of keys and values from off-chip High Bandwidth Memory (HBM) to on-chip Static Random Access Memory (SRAM), performs dequantization and the attention computations on chip, and writes partial outputs. Because the decoding phase exposes limited parallelism, we parallelize across KV blocks: each block produces a partial sum of the attention output in parallel, the partial sums are reduced to form the final attention output, and we then apply the KVLinC linear correction. Before running attention, we compute the linear-correction states $C_n$ and $C_d$ for the numerator and denominator, respectively: $C_n = \phi_q(Q_n)S_n, C_d = \phi_q(Q_n)P_n$. These states, together with the (de)quantized attention operands, are passed to the decoding algorithm in Algorithm 1. Following KIVI Liu et al. (2024b), we quantize the KV cache only after the attention computation; consequently, the prefill phase remains floating-point and can be accelerated with FlashAttention itself.

## 4 EXPERIMENTS

In this section, we benchmark KVLinC against competitive baselines. First we provide results of the algorithm and then we provide end to end hardware efficiency improvements provided by KVLinC.

### 4.1 SETUP

**Models, tasks, datasets and baselines.** We evaluate KVLinC on the Llama-3 (Meta, 2024a;b; Touvron et al., 2023), Qwen-2.5 (Yang et al., 2024a), and Qwen-3 (Yang et al., 2025) model families, chosen to test robustness of linear correction adapters under different architectural settings (Qwen-2.5 uses bias in query/key projections; Qwen-3 applies layernorm after them). We compare against KIVI (Liu et al., 2024b), QuaRot (Ashkboos et al., 2024), ResQ (Saxena et al., 2024b), and Gear (Kang et al., 2024). Since ResQ and Gear retain portions of the KV cache in high precision, we align their design point with KVLinC's average precision: ResQ keeps $3.125\%$ of channels in 16-bit, and Gear uses rank-2 quantization error. All methods quantize the KV cache to 2-bits with group size 128, while storing the most recent 128 tokens in full precision. We evaluate both base and instruction-tuned models. For base models, we measure perplexity on Wikitext (Merity et al., 2016) (2k sequence length, autoregressive generation with compressed KV cache), exact match accuracy

Table 1: Results of base LLMs on Wikitext perplexity (2k sequence length), 5-shot GSM8K and BBH. Average KV cache precision is computed considering the scaling factors and zero points along with components used to compensate for quantization error. ↑ higher is better, ↓: lower is better. *Upper bound performance.

| Method | KV Cache | Llama-2-7B | | | Llama-3.2-3B | | | Llama-3.1-8B | | |
|--------|----------|-----------|--------|------|-------------|--------|------|-------------|--------|------|
| | | Wikitext↓ | GSM8K↑ | BBH↑ | Wikitext↓ | GSM8K↑ | BBH↑ | Wikitext↓ | GSM8K↑ | BBH↑ |
| FP16* | 16 | 5.5 | 14.3 | 39.9 | 7.8 | 25.6 | 47.0 | 6.2 | 49.7 | 62.7 |
| KIVI | 2.46 | 5.9 | 10.6 | 30.5 | 11.0 | 11.8 | 25.0 | 7.8 | 34.1 | 44.2 |
| Quarot | 2.46 | 5.8 | 9.8 | 29.4 | 9.7 | 9.9 | 21.5 | 7.3 | 26.8 | 34.3 |
| ResQ | 2.91 | **5.7** | 10.8 | **32.6** | **8.7** | 14.1 | 31.5 | **6.8** | 36.2 | 42.7 |
| Gear-L | 2.96 | 5.8 | 10.8 | 30.0 | 10.0 | **16.4** | 28.3 | 7.3 | 38.8 | 46.7 |
| KVLinC | 2.71 | **5.7** | **11.0** | 31.1 | 9.4 | **16.4** | **32.7** | 7.1 | **40.9** | **48.6** |

| Method | KV Cache | Qwen2.5-1.5B | | | Qwen2.5-3B | | | Qwen2.5-7B | | |
|--------|----------|-----------|--------|------|-------------|--------|------|-------------|--------|------|
| | | Wikitext↓ | GSM8K↑ | BBH↑ | Wikitext↓ | GSM8K↑ | BBH↑ | Wikitext↓ | GSM8K↑ | BBH↑ |
| FP16* | 16 | 9.3 | 61.5 | 43.9 | 8.0 | 69.4 | 55.1 | 6.8 | 81.1 | 69.4 |
| KIVI | 2.46 | 16.5 | 26.9 | 17.9 | 9.7 | 46.1 | 32.7 | 11.2 | 71.0 | 45.3 |
| Quarot | 2.46 | 7268.2 | 0.1 | 0.0 | 783.0 | 0.0 | 0.0 | 3380.0 | 0.1 | 0.0 |
| ResQ | 2.91 | 13.2 | 10.6 | 22.1 | 9.1 | 47.2 | 39.2 | 10.6 | 35.6 | 47.9 |
| Gear-L | 2.96 | 14.0 | 32.2 | 21.7 | 9.3 | 47.4 | 34.1 | 10.6 | **71.8** | 49.5 |
| KVLinC | 2.71 | **13.0** | **36.3** | **23.6** | **8.9** | **47.6** | **35.3** | **10.5** | 71.2 | **50.1** |

| Method | KV Cache | Qwen3-1.7B-Base | | | Qwen3-4B-Base | | | Qwen3-8B-Base | | |
|--------|----------|-----------|--------|------|-------------|--------|------|-------------|--------|------|
| | | Wikitext↓ | GSM8K↑ | BBH↑ | Wikitext↓ | GSM8K↑ | BBH↑ | Wikitext↓ | GSM8K↑ | BBH↑ |
| FP16* | 16 | 9.4 | 69.3 | 53.2 | 7.9 | 76.0 | 71.3 | 7.0 | 82.3 | 77.3 |
| KIVI | 2.46 | 11.2 | 48.4 | 30.5 | 9.1 | 67.5 | 49.9 | 7.7 | 78.6 | 58.6 |
| Quarot | 2.46 | 1963.3 | 0.0 | 0.0 | 755.3 | 0.1 | 0.0 | 202.3 | 17.5 | 20.8 |
| ResQ | 2.9 | 12.2 | 20.4 | 18.8 | 9.0 | 48.9 | 51.0 | 7.8 | 71.7 | 58.5 |
| Gear-L | 2.96 | 10.7 | 47.5 | 33.2 | 8.8 | 66.9 | 55.1 | 7.6 | 78.6 | **63.2** |
| KVLinC | 2.71 | **10.4** | **53.9** | **35.5** | **8.6** | **67.6** | **55.2** | **7.5** | **78.9** | 61.7 |

Table 2: Results of Instruct LLMs on long context and instruction following tasks. Taskwise accuracy can be found in Appendix A.2,A.3.*Upper bound performance.

| Model | Method | KV Cache | RULER | | LongBench | IF-Eval | |
|-------|--------|----------|------|------|-----------|-------------|---------------|
| | | | 4k | 8k | | inst-strict | prompt-strict |
| Llama-3.2-3B-Instruct | FP16* | 16 | 92.5 | 88.1 | 40.4 | 79.3 | 71.2 |
| | KIVI | 2.46 | 76.7 | 70.3 | **39.4** | 74.6 | 64.9 |
| | KVLinC | 2.71 | **80.8** | **73.6** | **39.4** | **76.3** | **67.5** |
| Qwen-2.5-3B-Instruct | FP16* | 16 | 90.3 | 85.0 | 31.4 | 68.9 | 58.8 |
| | KIVI | 2.46 | 49.5 | 41.0 | 28.0 | 62.7 | 52.5 |
| | KVLinC | 2.71 | **60.9** | **51.1** | **28.2** | **66.0** | **56.8** |
| Qwen-3-4B-Instruct | FP16* | 16 | 92.7 | 88.6 | 31.9 | 47.6 | 33.6 |
| | KIVI | 2.46 | 83.7 | 79.9 | **31.2** | 44.8 | 31.8 |
| | KVLinC | 2.71 | **86.2** | **82.4** | 31.0 | **45.7** | **32.5** |

on 5-shot GSM8K (Cobbe et al., 2021), and average accuracy on Big-Bench Hard (BBH) (Suzgun et al., 2022). For instruction-tuned models, we report results on long-context benchmarks: RULER (Hsieh et al., 2024), LongBench (Bai et al., 2023), and IF-Eval (Zhou et al., 2023). LongBench follows the setup in KIVI, while other benchmarks use the lm-evaluation-harness (Gao et al., 2024).

**Implementation Details** We implement KVLinC in PyTorch (Paszke et al., 2019) using Hugging-Face Transformers (Wolf et al., 2020). We set rank of correction adapters as $D = 256$, adding $< 1\%$ extra parameters to the LLM. For base models, adapters are trained on Alpaca dataset (Taori et al., 2023) using Adam (Kingma & Ba, 2017) optimizer with learning rate 0.01, sequence length 3k, batch size 24, for 500 steps. For instruction-tuned models, training uses RedPajama dataset (Weber et al., 2024), sequence length 8k, batch size 8, for 1500 steps with Adam optimizer. Training Llama-3.1-8B on Alpaca takes 2 hours, and Llama-3.2-3B on RedPajama takes 11 hours on 4×NVIDIA H200 GPUs.

## 4.2 MAIN RESULTS

**Results on Base Models.** We evaluate the base LLMs of various sizes of Llama, Qwen-2.5, and Qwen-3 model families on perplexity (PPL) on Wikitext at 2k sequence length, 5-shot GSM8K, and BBH benchmark. The results are presented in Table 1. KVLinC manages to outperform or match the performance of strong baselines at lower KV cache bitwidth. Compared to Gear, KVLinC achieves

Table 3: Wikitext perplexity at 1-bit KV cache quantization. Quantization group size is kept at 64 and recent 128 tokens are kept in floating point.

| Method | KV cache | Qwen2.5-1.5B | Llama-3.2-3B | Llama-3.1-8B |
|---|---|---|---|---|
| | | Wikitext ↓ | Wikitext ↓ | Wikitext ↓ |
| KIVI | 1.73 | 242.6 | 287.1 | 204.9 |
| Gear-L | 1.92 | 97.3 | 164.7 | 98.2 |
| ResQ | 1.92 | 90.2 | 70.2 | 65.3 |
| KVLinC | 1.92 | **81.0** | **65.7** | **60.8** |
| Method | KV cache | Qwen3-1.7B-Base | Qwen3-4B-Base | Qwen3-8B-Base |
| | | Wikitext ↓ | Wikitext ↓ | Wikitext ↓ |
| KIVI | 1.73 | 143.9 | 45.4 | 40.2 |
| Gear-L | 1.92 | 147.3 | 39.6 | 36.6 |
| ResQ | 1.92 | 80.2 | 39.8 | 38.2 |
| KVLinC | 1.92 | **24.5** | **19.0** | **15.1** |

Table 4: Evaluation of KVLinC quantization in combination with $H_2O$ KV cache sparsification. Table shows Wikitext perplexity (lower is better) at varying KV cache sparsity levels.

| Model | Method | KV cache sparsity | | | | | |
|---|---|---|---|---|---|---|---|
| | | 70% | 75% | 80% | 85% | 90% | 95% |
| Qwen-3-1.7B-Base | KIVI | 11.6 | 11.7 | 11.9 | 12.2 | 12.8 | 14.1 |
| | KVLinC | **11.1** | **11.2** | **11.4** | **11.7** | **12.2** | **13.4** |
| Qwen-3-4B-Base | KIVI | 9.2 | 9.3 | 9.4 | 9.6 | 9.9 | 10.8 |
| | KVLinC | **8.9** | **9.0** | **9.1** | **9.2** | **9.5** | **10.3** |
| Qwen-3-8B-Base | KIVI | 7.8 | 7.8 | 7.9 | 8.0 | 8.2 | 8.8 |
| | KVLinC | **7.6** | **7.7** | **7.7** | **7.8** | **8.0** | **8.6** |

upto $6.4\%$ improvements on GSM8K and upto $2.3\%$ improvements on BBH benchmark. Greater improvements are observed for smaller-sized models. For the Qwen-2.5 and Qwen-3 family of models, QuaRot fails to produce meaningful results, showcasing that per token quantization strategy for both keys and values is sub-optimal. ResQ adopts the same quantization configuration as QuaRot but keeps important channels in high precision, enabling improved results. Since calibration for ResQ is done on Wikitext itself, it achieves surprisingly low Wikitext PPL on Llama models. KVLinC instead involves calibration on out-of-domain Alpaca dataset and does not overfit to any evaluation benchmarks.

**Results at 1-bit KV cache.** We also evaluate methods under extreme 1-bit KV-cache quantization, using a group size of 64 and keeping the rest of the setup unchanged. Table 3 reports Wikitext perplexity (lower is better) at a 2K sequence length. All techniques show a significant perplexity drop at 1-bit; however, KVLinC still outperforms all baselines. Notably, on Qwen-3-1.7B, KVLinC achieves 56 lower perplexity than the next best method, ResQ Saxena et al. (2024b).

**Results on Instruct models.** We evaluate the instruction tuned LLMs of Llama-3.1, Qwen-2.5 and Qwen-3 model families on RULER (4k and 8k sequence length), LongBench and IF-eval benchmarks. The results are presented in Table 2. KVLinC outperforms KIVI on all the presented models on RULER and IF-eval tasks. For the Qwen-2.5-3B instruct model, KVLinC achieves more than $10\%$ improvement on RULER tasks and upto $4.3\%$ on IF-eval tasks. For LongBench, quantization of KV cache impacts final accuracy by a small amount and the performance of both KIVI and KVLinC is comparable.

**Interaction with KV cache sparsification.** KV cache sparsification is another widely used method to reduce memory cost Zhang et al. (2023); Adnan et al. (2024). KVLinC is compatible with such pruning strategies and can be combined with them for additional compression. To demonstrate this, we pair $H_2O$ Zhang et al. (2023) with KVLinC: we train the adapter normally (without pruning), and during inference keep only the $H_2O$-selected tokens at 2-bit precision. The adapters then track quantization error only for the retained tokens. As shown in Table 4, KVLinC consistently outperforms KIVI across all sparsity levels, even up to $95\%$.

## 4.3 ANALYSIS

**Impact of different components.** Further, we analyse how the complementary strategies presented in KVLinC perform in isolation. To achieve this, we apply the linear correction states

Table 5: Performance with applying Hadamard rotation and linear correction in isolation on Llama family. ↑ higher is better, ↓: lower is better.

| Model | Method | Wikitext↓ | GSM8K↑ |
|---|---|---|---|
| 3.1-8B | KIVI | 7.8 | 34.1 |
| | KIVI + LinC | 7.3 | 38.4 |
| | $Q_C(\boldsymbol{K}), Q_T(\boldsymbol{VH})$ | 7.2 | 36.9 |
| | KVLinC | 7.1 | 40.9 |
| 3.2-3B | KIVI | 11.0 | 11.8 |
| | KIVI + LinC | 9.8 | 14.5 |
| | $Q_C(\boldsymbol{K}), Q_T(\boldsymbol{VH})$ | 9.7 | 13.9 |
| | KVLinC | 9.4 | 16.4 |

Table 6: Impact on wikitext PPL with applying KVLinC to different decoder layer blocks. Applying KVLinC to earlier decoder layers provides greater improvements.

| KVLinC | Improvement over KIVI (%) | |
|---|---|---|
| Layers | Qwen-2.5-1.5B | Qwen-3-1.7B-Base |
| [0-9] | 7.96 | 3.03 |
| [9-18] | 4.35 | 2.44 |
| [18-27] | 2.73 | 1.20 |
| [0-13] | 10.55 | 3.75 |
| [14-27] | 4.29 | 2.33 |
| [0-27] | 16.82 | 5.27 |

to KIVI (KIVI+LinC) and compare with a baseline which does channel-wise quantization on raw keys and token-wise quantization on hadamard rotated values. The results are presented in Table 5. For both Llama-3.1-8B and Llama-3.2-3B, applying linear correction provides improvements in Wikitext perplexity and 5-shot GSM8K accuracy. Similarly, opting for Hadamard based quantization for values improves performance over KIVI. While combining the two complementary techniques enables KVLinC to achieve further gains in performance.

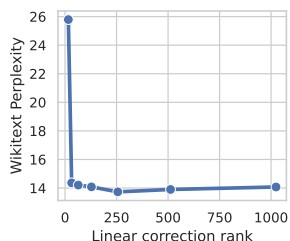

**Layerwise insights.** To better understand where KVLinC provides the most benefit, we selectively apply it to different subsets of decoder layers while using KIVI's quantization strategy for the remaining layers. On Qwen-2.5-1.5B and Qwen-3-1.7B-Base (both with 28 decoder layers), we observe that applying KVLinC to earlier layers yields greater improvements than applying it to the same number of later layers. As shown in Table 6, the Wikitext perplexity improve-

Figure 6: Linear correction rank ($D$) vs. perplexity.

ments (relative to KIVI) are consistently higher when KVLinC is applied to the initial layers. For example, applying KVLinC to the first 10 decoder layers achieves an average $3.5\%$ improvement over applying it to the last 10 layers. This finding highlights a key insight: the initial decoder layers play a more critical role under KV cache quantization.

**Dimension of Linear correction states.** The rank of the linear correction states $D$ controls the representational capacity of the feature maps, but higher ranks also increase overhead. As shown in Figure 6, Wikitext perplexity improves with larger ranks up to $D = 256$, beyond which gains saturate. We therefore select $D = 256$ as the optimal balance between accuracy and efficiency.

**Impact of Calibration data** We evaluate the sensitivity of KVLinC's adapters to different calibration datasets. For this analysis we evaluate KVLinC's downstream performance after calibration on three datasets : Alpaca Taori et al. (2023), LongAlpaca Chen et al. (2023) and C4 Dodge et al. (2021). As shown in Table 9, we find no clear consensus on the optimality of one particular dataset. The performance results for different datasets show no significant fluctuations.

### 4.4 HARDWARE SPEEDUP

We evaluate the end-to-end speedup of KVLinC to highlight the combined impact of KV cache quantization and our custom compute kernel. Specifically, we benchmark Llama-2-7B and Llama-3.1-8B using a prompt length of 256 tokens and generating 1024 output tokens, progressively increasing the batch size. Experiments are conducted on a single NVIDIA A40 (48 GB) GPU, measuring both memory usage and throughput (tokens per second). We compare KVLinC against FlashAttention-2 Dao (2023) with a 16-bit floating-point KV cache. As shown in Figure 7, quantizing the KV cache enables significantly larger batch sizes without exhausting memory. In particular, KVLinC supports up to $3.1\times$ more requests on Llama-3.1-8B and $3.5\times$ more requests on Llama-2-7B. Moreover, for Llama-2-7B, KVLinC delivers up to $2.55\times$ faster inference at batch size 32, beyond which FlashAttention becomes infeasible due to out-of-memory errors. For Llama-3.1-8B, the gains are more modest, with KVLinC achieving $1.2\times$ speedup at batch size 144. This discrepancy arises from ar-

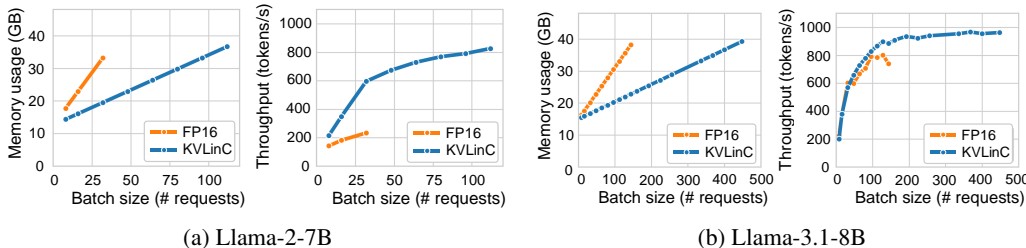

(a) Llama-2-7B                    (b) Llama-3.1-8B

Figure 7: End to end memory usage and throughput (tokens/s) on NVIDIA-A40 with varying batch sizes at prompt length of $256$ and $1024$ generated tokens for (a) Llama-2-7B and (b) Llama-3.1-8B.

chitectural differences: unlike Llama-3.1-8B, Llama-2-7B does not employ grouped query attention (GQA), resulting in a substantially larger KV cache that amplifies the benefits of our method.

## 5 RELATED WORKS

**KV Cache Quantization** The dynamic nature of KV caching introduces unique challenges for quantization, where both quantization and dequantization speed is critical. A variety of strategies have been explored across different granularities. ZipCache (He et al., 2024) and WKVQuant (Yue et al., 2024) adopt channel-separable, token-wise quantization, while KIVI (Liu et al., 2024b) applies channel-wise quantization to keys and token-wise quantization to values. In contrast, KVQuant (Hooper et al., 2024) and PolarQuant (Han et al., 2025) use non-linear quantization schemes to reduce error. QJL (Zandieh et al., 2025) introduces a specialized Johnson–Lindenstrauss transform for key tensors combined with per-token quantization of values. Other methods combine quantization with decomposition: Palu (Chang et al., 2024) and EigenAttention (Saxena et al., 2024a) integrate low-rank factorization with quantization. Several approaches further mitigate quantization error by leveraging advanced transformations or error modeling. QuaRot (Ashkboos et al., 2024) and Spin-Quant (Liu et al., 2024a) use Hadamard transforms to improve quantization robustness. ResQ (Saxena et al., 2024b) preserves salient channels in higher precision, whereas GEAR (Kang et al., 2024) maintains a low-rank approximation of the quantization error. Finally, MiKV (Yang et al., 2024b), QAQ (Dong et al., 2024), and SKVQ (Duanmu et al., 2024) explore variable bit-width schemes to balance accuracy with memory savings.

**Linear Attention** A large body of prior work has explored more efficient sequence modeling modules as alternatives to softmax attention in transformers, often by pretraining architectures from scratch. Within this line, numerous linear attention approaches have been proposed Choromanski et al. (2020); Katharopoulos et al. (2020); Xiong et al. (2021); Yang et al. (2023). More recently, several efforts focus on post-training conversion of softmax-attention transformers into linear counterparts. For example, Lolcats (Zhang et al., 2024) employs advanced linear feature map design combined with attention distillation, while Liger (Lan et al., 2025) incorporates gated recurrence to achieve this transition. Pushing further, LoLA (McDermott et al., 2025) and Based (Arora et al., 2025) adopt hybrid strategies that combine linear attention with selective application of exact softmax attention on subsets of keys and values, thereby improving accuracy while retaining efficiency.

## 6 CONCLUSION

In this work, we introduced *KVLinC*, a framework designed to mitigate attention errors arising from KV cache quantization. KVLinC integrates two complementary techniques to enable robust low-precision caching. First, through a detailed analysis of Hadamard rotation based quantization strategies, we showed that applying channel-wise quantization to raw keys and token-wise quantization to Hadamard-transformed values minimizes quantization error. Second, to address residual errors from quantized keys, we proposed lightweight linear correction adapters that explicitly learn to compensate for distortions in attention. Extensive evaluation across the Llama, Qwen2.5, and Qwen3 model families demonstrates that KVLinC consistently matches or surpasses strong baselines under aggressive KV-cache compression. Finally, we developed a custom attention kernel that delivers up to $2.55\times$ speedup over FlashAttention, enabling scalable, efficient, and long-context LLM inference.

REPRODUCIBILITY STATEMENT

We have provided details about our proposed algorithm in Section 4.1. Additionally, we provide codebase to reproduce results of our experiments and the baselines in supplementary materials.

ACKNOWLEDGMENTS

The authors would like to thank Sakshi Choudhary and Manish Nagaraj for helpful discussions. This work was supported by the Center for the Co-Design of Cognitive Systems (COCOSYS), a DARPA sponsored JUMP center of Semiconductor Research Corporation (SRC), Intel, SRC AIHW Program.

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

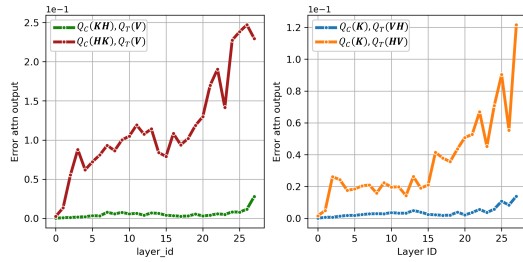

Figure 8: Attention error with pre- and post- Hadamard multiplication.

Amir Zandieh, Majid Daliri, and Insu Han. Qjl: 1-bit quantized jl transform for kv cache quantization with zero overhead. In *Proceedings of the AAAI Conference on Artificial Intelligence*, volume 39, pp. 25805–25813, 2025.

Michael Zhang, Simran Arora, Rahul Chalamala, Alan Wu, Benjamin Spector, Aaryan Singhal, Krithik Ramesh, and Christopher Ré. Lolcats: On low-rank linearizing of large language models. *arXiv preprint arXiv:2410.10254*, 2024.

Zhenyu Zhang, Ying Sheng, Tianyi Zhou, Tianlong Chen, Lianmin Zheng, Ruisi Cai, Zhao Song, Yuandong Tian, Christopher Ré, Clark Barrett, et al. H2o: Heavy-hitter oracle for efficient generative inference of large language models. *Advances in Neural Information Processing Systems*, 36:34661–34710, 2023.

Jeffrey Zhou, Tianjian Lu, Swaroop Mishra, Siddhartha Brahma, Sujoy Basu, Yi Luan, Denny Zhou, and Le Hou. Instruction-following evaluation for large language models. *arXiv preprint arXiv:2311.07911*, 2023.

# A APPENDIX

## A.1 ATTENTION ERROR WITH PRE- AND POST- HADAMARD MULTIPLICATION.

Figure 8 shows mean squared error in attention output under pre-multiplication and post-multiplication of Hadamard rotation. pre-multiplication mixes tokens prior to quantization, which amplifies quantization noise and injects errors into the attention logits. In contrast, post-multiplication only rotates channels within each token, thereby preserving relative token alignment and resulting in significantly more stable performance.

## A.2 DETAILED RESULTS ON LONGBENCH TASKS

Here we show task-wise accuracy on various tasks within the LongBench benchmark Bai et al. (2023). The results are presented in Table 7. We evaluate on 14 english language tasks on Long-Bench. Both KIVI and KVLinC show comparable performance on various tasks.

Table 7: Taskwise accuracy on LongBench tasks. [*]Upper bound performance.

| Model | Method | KV Cache | LongBench Tasks | | | | | | | | | | | | | | |
|---|---|---|---|---|---|---|---|---|---|---|---|---|---|---|---|---|---|
| | | | Multi News | Passage Count | Samsum | MFQA | Narrative QA | Hotpot QA | Trec | Qmsum | Trivia QA | Qasper | 2Wiki Mqa | Musique | Gov Report | Passage Retrieval | Avg. |
| Llama-3.2-3B-Instruct | FP16 | 16 | 26.2 | 3.5 | 42.5 | 51.1 | 26.3 | 30.3 | 71.0 | 22.7 | 88.9 | 40.6 | 28.0 | 13.7 | 33.5 | 86.8 | 40.4 |
| | KIVI | 2.46 | 24.7 | **3.6** | 40.7 | **48.5** | **27.1** | **30.6** | 70.5 | 23.9 | **88.7** | 36.1 | **32.5** | 13.9 | 26.4 | 83.7 | **39.4** |
| | KVLinC | 2.71 | **26.2** | 2.5 | 41.7 | 47.6 | 26.7 | 29.0 | 70.5 | 24.2 | 87.6 | 36.3 | 31.4 | 14.4 | 29.9 | 84.3 | 39.4 |
| Qwen-2.5-3B-Instruct | FP16 | 16 | 24.8 | 3.0 | 44.2 | 38.7 | 10.9 | 19.9 | 68.5 | 23.4 | 87.1 | 16.4 | 15.2 | 12.4 | 32.4 | 42.8 | 31.4 |
| | KIVI | 2.46 | **23.5** | **3.0** | **42.2** | 28.1 | 9.2 | **18.3** | 68.0 | **24.3** | 85.6 | 11.4 | **13.2** | **9.7** | 24.3 | 31.2 | 28.0 |
| | KVLinC | 2.71 | 23.0 | 2.2 | 41.6 | **31.0** | **10.1** | 14.2 | 68.0 | 24.1 | **86.6** | **12.1** | 13.2 | 9.0 | **27.3** | **31.8** | **28.2** |
| Qwen-3-4B-Instruct | FP16 | 16 | 19.8 | 3.3 | 44.1 | 24.8 | 3.5 | 13.4 | 73.0 | 23.4 | 88.8 | 11.1 | 14.4 | 10.0 | 29.2 | 88.0 | 31.9 |
| | KIVI | 2.46 | **22.9** | **4.6** | **42.0** | 21.3 | 3.7 | **13.1** | 73.0 | 22.4 | 87.8 | 10.7 | **13.6** | 7.9 | 26.2 | **87.5** | 31.2 |
| | KVLinC | 2.71 | 22.7 | 3.2 | 41.9 | **21.8** | **4.3** | 12.8 | 73.0 | 23.2 | **88.3** | 11.0 | 12.9 | **8.9** | **27.1** | 83.6 | 31.0 |

Table 8: Task-wise accuacy on RULER benchmark[*]Upper bound performance.

| Model | Seq-len | Method | KV Cache | RULER Tasks | | | | | | | | | | | | | |
|---|---|---|---|---|---|---|---|---|---|---|---|---|---|---|---|---|---|
| | | | | niahm1 | niahm2 | niahm3 | niah multiq | niah mltiV | niahs1 | niahs2 | niahs3 | cwe | fwe | hotpotqa | squadqa | vt | Avg. |
| Llama-3.2-3B-Instruct | 4k | FP16 | 16 | 99.8 | 100.0 | 98.4 | 100.0 | 99.8 | 100.0 | 100.0 | 99.6 | 95.1 | 93.1 | 55.2 | 68.9 | 92.1 | 92.5 |
| | | KIVI | 2.46 | **98.8** | 89.0 | 20.2 | 91.6 | 90.6 | **99.0** | **99.6** | 53.0 | 79.2 | 86.5 | **52.0** | 66.4 | 70.6 | 76.7 |
| | | KVLinC | 2.71 | 95.4 | **94.4** | **23.4** | **93.9** | **95.3** | 96.0 | 98.8 | **80.4** | **85.8** | **88.2** | 51.8 | **69.2** | **77.4** | **80.8** |
| | 8k | FP16 | 16 | 98.4 | 99.8 | 96.0 | 99.5 | 99.5 | 100.0 | 100.0 | 99.8 | 66.9 | 85.6 | 52.6 | 63.8 | 84.0 | 88.1 |
| | | KIVI | 2.46 | **97.0** | 82.8 | 5.6 | 90.1 | 92.0 | **99.4** | **99.0** | 55.2 | 38.9 | 74.3 | **51.2** | 56.3 | 72.0 | 70.3 |
| | | KVLinC | 2.71 | 93.0 | **92.6** | **10.2** | **92.7** | **92.5** | 93.6 | 96.8 | **73.2** | **50.0** | **80.9** | 49.4 | **59.8** | **72.4** | **73.6** |
| Qwen-2.5-3B-Instruct | 4k | FP16 | 16 | 99.8 | 99.0 | 97.4 | 100.0 | 99.5 | 100.0 | 84.7 | 99.8 | 84.7 | 91.5 | 49.0 | 72.1 | 96.6 | 90.3 |
| | | KIVI | 2.46 | 65.0 | 47.6 | 0.0 | 58.5 | 44.7 | 66.0 | 56.6 | 2.4 | **65.3** | 79.9 | **43.0** | 62.3 | 51.9 | 49.5 |
| | | KVLinC | 2.71 | **82.4** | **52.6** | **0.6** | **75.7** | **70.4** | **87.6** | **87.6** | **16.6** | 65.2 | **81.7** | 42.8 | **64.4** | **63.8** | **60.9** |
| | 8k | FP16 | 16 | 100.0 | 99.6 | 87.6 | 100.0 | 98.4 | 100.0 | 100.0 | 100.0 | 46.3 | 77.1 | 43.6 | 58.5 | 94.5 | 85.0 |
| | | KIVI | 2.46 | 57.8 | 26.4 | **0.0** | 55.3 | 39.7 | 69.0 | 56.6 | 3.8 | **34.2** | 61.4 | 34.8 | 45.3 | 49.6 | 41.1 |
| | | KVLinC | 2.71 | **75.8** | **34.6** | **0.0** | **69.9** | **64.4** | **85.6** | **81.2** | **21.6** | 29.6 | **65.3** | **36.4** | **48.0** | **52.4** | **51.1** |
| Qwen-3-4B-Instruct | 4k | FP16 | 16 | 97.4 | 100.0 | 99.8 | 99.6 | 98.3 | 100.0 | 100.0 | 99.8 | 94.7 | 88.1 | 54.8 | 72.1 | 100.0 | 92.7 |
| | | KIVI | 2.46 | 96.6 | 95.0 | 48.0 | 97.4 | 96.6 | **99.2** | 96.8 | 78.8 | **81.9** | 82.1 | 55.6 | 69.6 | 90.9 | 83.7 |
| | | KVLinC | 2.71 | **97.4** | **96.6** | **63.6** | **98.1** | **98.0** | 99.0 | **97.8** | **84.6** | 81.2 | **83.1** | **56.0** | **71.3** | **94.4** | **86.2** |
| | 8k | FP16 | 16 | 97.8 | 99.0 | 99.4 | 99.3 | 96.2 | 100.0 | 100.0 | 100.0 | 66.7 | 83.6 | 50.6 | 59.8 | 99.2 | 88.6 |
| | | KIVI | 2.46 | 96.2 | 91.2 | 26.4 | 96.6 | 96.4 | 99.2 | 96.0 | 75.4 | 67.8 | **82.1** | 55.4 | **63.7** | 92.2 | 79.9 |
| | | KVLinC | 2.71 | **97.8** | **94.2** | **42.2** | **98.2** | **97.7** | **99.8** | **96.6** | **83.2** | **69.5** | 79.8 | **55.6** | 63.1 | **93.1** | **82.4** |

Table 9: Downstream performance of KVLinC with different calibration datasets.

| Model | Dataset | Wiki ↓ | Gsm8k ↑ (em) | BBH ↑ |
|---|---|---|---|---|
| Llama-3.2-3B | Alpaca | 9.4 | 16.4 | 32.7 |
| | Long Alpaca | 9.6 | 16.4 | 32.5 |
| | C4 | 9.5 | 16.6 | 33 |
| Qwen2.3-3B | Alpaca | 8.9 | 47.6 | 35.2 |
| | Long Alpaca | 9 | 48 | 34.7 |
| | C4 | 8.7 | 46.4 | 35.4 |
| Qwen3-4B-Base | Alpaca | 8.6 | 67.6 | 55.2 |
| | Long Alpaca | 8.8 | 66.6 | 55.5 |
| | C4 | 8.7 | 65.4 | 54.7 |

## A.3 DETAILED RESULTS ON RULER TASKS

Additionally we also provide task wise breakdown in RULER benchmark in Table 8. The results are presented for both 4k and 8k sequence length. As shown in Table 8, KVLinC outperforms KIVI on most of the individual tasks across sequence lengths and models.

## A.4 IMPACT OF CALIBRATION DATA

Table 9, shows downstream performance of KVLinC with adapters trained using different calibration datasets. Table shows minimal performance variation, demonstrating the robustness of KVLinC's calibration

## A.5 LLM USAGE

The authors of this paper used ChatGPT (`https://chatgpt.com/`) for polishing text within this paper. The authors take full responsibility for the content within this paper.

