# OpenReview forum: "KVLinC: KV Cache Quantization with Hadamard Rotation and Linear Correction"
_ICLR.cc/2026/Conference — Submitted to ICLR 2026_

### Official Review · Reviewer_636Y · 2025-10-25

**Soundness:** 3
**Presentation:** 2
**Contribution:** 2
**Rating:** 4
**Confidence:** 4

**Summary:**

This work proposes a framework for KV-cache compression involving Hadamard rotation to simplify the quantization problem, combined with a lightweight correction adapter. The authors provide an ablation study on the choice of quantization axis and rotation, adopting the best-performing option. To compensate for quantization error, they train dedicated correction modules. The approach is validated on several transformer model families for both short and long context tasks.

**Strengths:**

* The proposed method achieves a favorable compression-performance trade-off, outperforming several known baselines from the literature.

* The method is accompanied by a custom attention kernel that speeds up inference.

**Weaknesses:**

* The overall idea of applying rotation followed by linear correction is not novel. The AQUA-KV [4] method also involves training predictors and cache rotation. While the predictor in this work is implemented differently, the core concept is similar.

* The ablation study on the choice of KV-cache quantization axis, despite being useful, is not novel and was previously explored in [1] and [2], where channel-wise key quantization was also found to be preferable.

* The comparisons in Tables 1 and 2 are not entirely fair. The bitwidth difference between methods may be up to **0.5 bits** on average, placing methods with smaller bitwidths at a disadvantage relative to those with higher bitwidths.

 * I suggest adding comparisons with additional baselines:
    * KVQuant [2] is an established cache quantization method
    * QJL [3] demonstrates strong performance at 3-bit compression and provides kernels for faster inference
    * AQUA-KV [4], being closest in terms of formulation, is a natural baseline

---
References

[1] Liu, Zirui, et al. "Kivi: A tuning-free asymmetric 2bit quantization for kv cache." arXiv preprint arXiv:2402.02750 (2024).

[2] Hooper, Coleman, et al. "Kvquant: Towards 10 million context length llm inference with kv cache quantization." Advances in Neural Information Processing Systems 37 (2024): 1270-1303.

[3] Zandieh, Amir, Majid Daliri, and Insu Han. "Qjl: 1-bit quantized jl transform for kv cache quantization with zero overhead." Proceedings of the AAAI Conference on Artificial Intelligence. Vol. 39. No. 24. 2025.

[4] Shutova, Alina, et al. "Cache Me If You Must: Adaptive Key-Value Quantization for Large Language Models." arXiv preprint arXiv:2501.19392 (2025).

**Questions:**

This work primarily targets KV cache compression of 2.71 bits per parameter on average. How well does the method perform under even stronger compression—around 2 bits or below?

---

> ### Author Response · Authors · 2025-11-25
> **Response to Reviewer 636Y**
>
> Thank you Reviewer 636Y for your constructive comments. Please find our response to your questions below:
>
> 1. **Analysis at stronger KV cache compression**
>
> We further analyze quantization techniques at 1-bit KV cache quantization. The KV cache quantization hyperparameters include group size of 64 with 128 recent tokens kept in full precision. The results are presented in Table 1 below. As can be seen, under this extreme KV cache compression, all techniques exhibit a significant increase in perplexity. However, KVLinC still maintains superior performance by achieving lowest perplexity compared with baselines.
>
> Table 1: Wikitext perplexity (lower is better) at 1-bit KV cache quantization.
>
> | Model            | Method | KV cache (bits) | Wikitext PPL |
> | ---------------- | ------ | --------------- | ------------ |
> | **Qwen2.5-1.5B** | KIVI   | 1.73            | 242.6        |
> |                  | Gear-L | 1.92            | 97.3         |
> |                  | ResQ   | 1.92            | 90.2         |
> |                  | KVLinC | 1.92            | **81.0**     |
> | **Llama-3.2-3B** | KIVI   | 1.73            | 287.1        |
> |                  | Gear-L | 1.92            | 164.7        |
> |                  | ResQ   | 1.92            | 70.2         |
> |                  | KVLinC | 1.92            | **65.7**     |
> | **Llama-3.1-8B** | KIVI   | 1.73            | 204.9        |
> |                  | Gear-L | 1.92            | 98.2         |
> |                  | ResQ   | 1.92            | 65.3         |
> |                  | KVLinC | 1.92            | **60.8**     |
> | **Qwen3-1.7B**   | KIVI   | 1.73            | 143.9        |
> |                  | Gear-L | 1.92            | 147.3        |
> |                  | ResQ   | 1.92            | 80.2         |
> |                  | KVLinC | 1.92            | **24.5**     |
> | **Qwen3-4B**     | KIVI   | 1.73            | 45.4         |
> |                  | Gear-L | 1.92            | 39.6         |
> |                  | ResQ   | 1.92            | 39.8         |
> |                  | KVLinC | 1.92            | **19.0**     |
> | **Qwen3-8B**     | KIVI   | 1.73            | 40.2         |
> |                  | Gear-L | 1.92            | 36.6         |
> |                  | ResQ   | 1.92            | 38.2         |
> |                  | KVLinC | 1.92            | **15.1**     |
>
> 2. **Comparison with more baselines**
>
> We provide comparison with KVQuant [1] and AquaKV [2] in Table 2 below. Unfortunately, we were unable to run QJL [3] baseline. Among the presented baselines, KVQuant utilizes non linear quantization and keeps 1% channels in full precision. AquaKV uses previous layer keys and values to predict current keys and values and only stores 2-bit quantized prediction error in KV cache. AquaKV uses simple linear regression to predict next layer keys and values, therefore, leads to a drastic decline in downstream performance when the prediction deviates substantially from ground truth. Since KVLinC uses non-linear prediction coupled with gradient based learning, it maintains more stable performance across model architectures. KVLinC always achieves better results compared with KVQuant. Compared with AquaKV, KVLinC either matches performance or surpasses AquaKV at 2-bit quantization.
>
> Table 2 : Wikitext perplexity (lower is better) at 2-bit KV cache quantization.
>
> | Model        | Aqua-kv-higgs | KVQuant | KVLinC   |
> | ------------ | ------------- | ------- | -------- |
> | Llama-2-7b   | 634.2         | 6.0     | **5.7**  |
> | Llama-3.2-3B | **8.1**       | 9.8     | 9.4      |
> | Llama-3.1-8B | **6.5**       | 7.5     | 7.1      |
> | Qwen2.5-1.5b | 6238.0        | 11.4    | **13**   |
> | Qwen2.5-3b   | **8.7**       | 9.5     | 8.9      |
> | Qwen2.5-7b   | 23187.0       | 11.0    | **10.5** |
>
> [1] Hooper, Coleman, et al. "Kvquant: Towards 10 million context length llm inference with kv cache quantization." _Advances in Neural Information Processing Systems_ 37 (2024): 1270-1303.
>
> [2] Shutova, Alina, et al. "Cache Me If You Must: Adaptive Key-Value Quantization for Large Language Models." _arXiv preprint arXiv:2501.19392_ (2025).

---

### Official Review · Reviewer_nuEP · 2025-10-28

**Soundness:** 3
**Presentation:** 3
**Contribution:** 2
**Rating:** 4
**Confidence:** 4

**Summary:**

This paper addresses the challenge of KV cache compression in large language model (LLM) inference, where key–value tensors dominate memory and bandwidth costs as context length grows.
The authors propose KVLinC, a method that combines (1) Hadamard rotation for distribution smoothing and (2) linear correction adapters to mitigate quantization-induced bias in the attention mechanism.

**Strengths:**

1. Empirical thoroughness: Evaluated on multiple model families, sizes, and long-context benchmarks (RULER, GSM8K, LongBench)
2. Strong performance at extremely low precision (2-bit): Outperforms several state-of-the-art quantization baselines across diverse tasks.

**Weaknesses:**

1. Ablation clarity: The paper would benefit from explicit comparisons between “rotation only,” “correction only,” and “both” configurations on the same setup.
2. The contribution is incremental, as both Hadamard rotation and linear correction have been introduced in prior work; this paper mainly demonstrates an effective combination of the two techniques.

**Questions:**

1. How does KVLinC interact with KV cache sparsification or compression-based pruning?
2. Furthuer ablation study needed, See weakness1.

---

> ### Author Response · Authors · 2025-11-25
> **Response to Reviewer nuEP**
>
> Thank you Reviewer nuEP for your constructive comments. We are delighted to hear your appreciation about the thoroughness and strong empirical performance of KVLinC. Please find our response to your questions below:
>
> 1. **Ablation Clarity**
>
> The iso-setting comparison between "rotation-only", "correction-only" and "both" signifying the impact of different components of KVLinC is already provided in Table 3 of our manuscript. In the Table, "KIVI + LinC" baseline denotes "correction-only" baseline, "$Q_{C}(K),\; V_{T}(VH)$" denotes "rotation-only" baseline while "KVLinC" denotes the combination of both techniques. We apologize for any misunderstanding and we will clarify in the paper appropriately.
>
> 2. **Impact of KV cache sparsification**
>
> This is a very helpful suggestion. KVLinC is indeed compatible with KV cache sparsification approaches and incorporating this would improve the value of our work. As a case study, we leverage $H_2O$ [1] KV cache sparsification approach and analyze downstream task performance at different sparsity levels. The results are provided in Table 1 below. We compare KVLinC and KIVI [2] at 2-bit quantization at varying KV cache sparsity levels. The key and value tokens which are not pruned are quantized to 2-bit using respective approaches. As showing in Table 1, KVLinC achieves better performance (lower Wikitext perplexity) compared with KIVI across all sparsity levels.
>
>
>
> Table 1: Perplexity on Wikitext (lower is better) when $H_2O$ KV cache sparsification is applied over kv cache quantization.
>
> | Model           | Method          |         | **KV cache Sparsity** |         |         |         |         |
> | --------------- | --------------- | ------- | --------------------- | ------- | ------- | ------- | ------- |
> |                 |                 | **70%** | **75%**               | **80%** | **85%** | **90%** | **95%** |
> | **Qwen-3-1.7B** | KIVI + $H_2O$   | 11.56   | 11.71                 | 11.91   | 12.23   | 12.82   | 14.12   |
> |                 | KVLinC + $H_2O$ | 11.06   | 11.21                 | 11.38   | 11.68   | 12.20   | 13.41   |
> | **Qwen-3-4B**   | KIVI + $H_2O$   | 9.21    | 9.28                  | 9.38    | 9.56    | 9.89    | 10.76   |
> |                 | KVLinC + $H_2O$ | 8.89    | 8.96                  | 9.05    | 9.22    | 9.54    | 10.31   |
> | **Qwen-3-8B**   | KIVI + $H_2O$   | 7.80    | 7.83                  | 7.88    | 7.97    | 8.18    | 8.75    |
> |                 | KVLinC + $H_2O$ | 7.63    | 7.66                  | 7.71    | 7.81    | 8.01    | 8.56    |
>
>
> [1] Zhang, Zhenyu, et al. "H2o: Heavy-hitter oracle for efficient generative inference of large language models." _Advances in Neural Information Processing Systems_ 36 (2023): 34661-34710.
>
> [2] Liu, Zirui, et al. "Kivi: A tuning-free asymmetric 2bit quantization for kv cache." _arXiv preprint arXiv:2402.02750_ (2024).

---

### Official Review · Reviewer_WeNX · 2025-10-30

**Soundness:** 3
**Presentation:** 3
**Contribution:** 2
**Rating:** 6
**Confidence:** 3

**Summary:**

- Asymmetric Hadamard rotation and quantization. The authors analyze several rotation and axis configurations and find that quantizing raw keys channel-wise and Hadamard-rotated values token-wise yields the best trade-off. This differs from QuaRot, which apply Hadamard rotations to both keys and values before token-wise quantization. I believe the rationale is that keys exhibit heavy-tailed channel distributions that are sensitive to rotation, while values benefit from rotation.
- Linear correction adapters. Lightweight trainable modules are added to compensate for distortions in the attention weights induced by quantized keys. These adapters add <1% parameters and introduce constant memory overhead, scaling linearly with sequence length.
- A custom Triton kernel integrates quantization and correction for efficient decoding. Experiments across Llama-3, Qwen-2.5, and Qwen-3  and KIVI, Gear, and ResQ on short- and long-context benchmarks (Wikitext, GSM8K, BBH, RULER, LongBench).

**Strengths:**

- Clear motivation and strong practical relevance. The paper addresses a key bottleneck in LLM inference, the KV-cache memory footprint, and focuses specifically on the challenging 2-bit quantization regime, which is highly relevant for deployment.
- Well-engineered and empirically thorough. I like that the authors implemented a dedicated Triton kernel to show that there are gains rather than just arguing the same, but failing against optimized benchmarks.
- Effective combination of existing ideas. Although both Hadamard rotation to KV and lightweight adapters are known concepts, their combination and tuning for extreme low precision is original in execution and yields consistently strong results.
- Clarity and reproducibility. The paper is clearly written, with helpful figures and detailed methodology.

**Weaknesses:**

- Limited novelty relative to prior work. The use of Hadamard rotation for quantization was already explored in QuaRot and TurboAttention, which both apply rotations to keys and values to mitigate outliers. KVLinC’s main advance is the asymmetric configuration (rotating only values and quantizing keys channel-wise), which is an empirical refinement rather than a new conceptual mechanism
- Correction adapters require retraining. The proposed linear correction mechanism is elegant but depends on fine-tuning small adapter modules on a calibration dataset. This breaks the plug-and-play nature of tuning-free quantizers such as KIVI or QuaRot, and may limit practicality in deployment across models or domains. Evaluating how sensitive the adapters are to the choice of calibration data or whether they can generalize across architectures would strengthen the claim.

**Questions:**

- The implementation modify attention calculation, is the proposed approach compatible with modern attention implementation such as LeanAttention (https://arxiv.org/abs/2405.10480).
- is there a way of avoiding the training/fine-tuning, and what would be the accurcay impact? What is the sentisivity on selection of calibration dataset?
- Interaction with attention kernel optimization. The paper presents a custom Triton kernel for decoding. Have the authors looked at integrating it into existing frameworks?

---

> ### Author Response · Authors · 2025-11-25
> **Response to Reviewer WeNX**
>
> Thank you Reviewer WeNX for your constructive comments and for finding our research paper through and relevant. Please find our response to your concerns below.
>
> 1. **Compatibility with Lean Attention**
>
> Lean Attention [1] proposes an optimized attention compute flow without modifying the softmax attention operation itself. In principal, KVLinC is compatible with Lean Attention and requires modifying the proposed attention kernel to incorporate Lean Attention's compute flow proposed. For our kernel, we adopted Flash Decoding's compute flow which involves parallelizing attention over blocks of keys and values. Incorporating Lean Attention's stream-K style partitioning will help us further improve the efficiency of our kernel. We will explore this in our future work.
>
> 2. **Avoiding training/finetuning of adapters**
>
> Unfortunately, a key feature and possibly a limitation of our approach is the requirement of data driven finetuning. We were unable to obtain a parameterization of correction adapters which involves a closed form solution and is training free.
>
> 3. **Impact of calibration data**
>
> We further provide results on additional calibration datasets 1) Alpaca (originally used in KVLinC), 2) Long Alpaca[2], 3) C4 [3]. The results are provided below in Table 1. We find no clear consensus on the optimality of one particular dataset. The performance results for different datasets show no significant fluctuations.
>
>
> Table 1: Impact of calibration dataset on downstream tasks.
>
> | Model             | Dataset     | Wiki | GSM8k (em) | BBH  |
> | ----------------- | ----------- | ---- | ---------- | ---- |
> | **Llama-3.2-3B**  | Alpaca      | 9.4  | 16.4       | 32.7 |
> |                   | Long Alpaca | 9.6  | 16.4       | 32.5 |
> |                   | C4          | 9.5  | 16.6       | 33   |
> | **Qwen2.3-3B**    | Alpaca      | 8.9  | 47.6       | 35.2 |
> |                   | Long Alpaca | 9.0  | 48.0       | 34.7 |
> |                   | C4          | 8.7  | 46.4       | 35.4 |
> | **Qwen3-4B-Base** | Alpaca      | 8.6  | 67.6       | 55.2 |
> |                   | Long Alpaca | 8.8  | 66.6       | 55.5 |
> |                   | C4          | 8.7  | 65.4       | 54.7 |
>
>
> 4. **Integrating KVLinC to existing framework**
> Thank you for this valuable suggestion. We are currently working on integrating our framework with vLLM [4] to increase the visibility and adoption of our work.
>
> [1] Sanovar, Rya, et al. "Lean Attention: Hardware-Aware Scalable Attention Mechanism for the Decode-Phase of Transformers." _arXiv preprint arXiv:2405.10480_ (2024).
>
> [2] Chen, Yukang, et al. "Longlora: Efficient fine-tuning of long-context large language models." _arXiv preprint arXiv:2309.12307_ (2023).
>
> [3] Dodge, Jesse, et al. "Documenting large webtext corpora: A case study on the colossal clean crawled corpus." _arXiv preprint arXiv:2104.08758_ (2021).
>
> [4] https://github.com/vllm-project/vllm

---

### Official Review · Reviewer_8jyt · 2025-10-31

**Soundness:** 3
**Presentation:** 3
**Contribution:** 3
**Rating:** 6
**Confidence:** 4

**Summary:**

The paper introduces KVLinC, a framework for quantizing KV cache to 2-bit precision while mitigating attention errors. It combines Hadamard rotation to reduce quantization errors in values with lightweight linear correction adapters that compensate for errors in quantized keys, achieving higher compression (using a calibration dataset) and performance than baselines like KIVI, QuaRot, ResQ, and Gear. Evaluations on LLaMA-3, Qwen-2.5, and Qwen-3 families show improved perplexity and accuracy on short and long-context benchmarks, plus a custom Triton kernel enabling up to 2.55× faster inference.

**Strengths:**

- The paper's originality lies in the hybrid approach: analyzing Hadamard rotation axes (observations 1-3, lines 189-211) to find an optimal config without extra compute overhead, and introducing trainable adapters (eq. 4-6, lines 216-257) that add <1% parameters but scale efficiently (O(dD) decode cost). Quality is high, with rigorous ablations (Fig. 3) and error reduction evidence (Fig. 5). Clarity is strong in explaining KV stats (Fig. 2, lines 108-161) and motivations (e.g., error accumulation in long sequences, lines 44-46). Significance is evident for efficient LLM deployment, as KV quantization enables larger batches/long contexts
- End-to-end systems story. A Triton kernel that streams bit-packed KV blocks and applies correction yields concrete throughput/batch-size gains on real models.
- Code is well-written and solid (reproducible).

**Weaknesses:**

- Missing SpinQuant baseline. Given the use of a calibration set (Alpaca for base and RedPajama for instruct), a fairer comparison should include SpinQuant (KV-only) at the same 2-bit setting, keeping weights/activations FP to isolate the KV effect. The paper compares to QuaRot (random Hadamard) but omits SpinQuant’s learned rotation.
- Minor: the evaluation scope is limited to smaller models (up to 8B).

**Questions:**

- In Sec. 3.1 (lines 189-194), Observation 1 states pre-multiplication with Hadamard yields worse performance due to token mixing; can you provide quantitative error metrics (e.g., MSE on attention logits) comparing pre- vs. post-multiplication on a specific model like Qwen-2.5-3B to support this?

- For linear correction adapters in eq. 5, how sensitive is performance to the feature map rank D=256 (line 320)? Could you report perplexity on Wikitext for D=128/512 to assess parameter overhead trade-offs?

- In Fig. 5, what is the corresponding downstream impact on a long-context benchmark like RULER (Sec. 4.1), e.g., accuracy drop without adapters?

---

> ### Author Response · Authors · 2025-11-25
> **Response to Reviewer 8jyt**
>
> Thank you Reviewer 8jyt for your constructive comments. We are delighted that you appreciate the significance of our work KVLinC. Please find our response to your comments below.
> 1. **Missing Spin Quant [1] baseline**
>
> Thank you for bringing up SpinQuant [1], a relevant work. Upon careful perusal of their approach, SpinQuant still uses Hadamard rotation for quantizing keys and values in KV cache. SpinQuant's learnable rotations approach is only applied to linear layer weights and activations while rotations for keys and values is kept to be Hadamard. Therefore, from the perspective of KV cache quantization, both Quarot [2] and SpinQuant are identical. Since we already compare with Quarot and demonstrate that KVLinC outperforms their approach, KVLinC would also outperform SpinQuant when considering KV cache quantization in islotation (with weights and activation kept in FP).
>
> 2. **Ablation on pre- and post- Hadamard multiplication**
>
> As requested, we provide results on attention output error between floating point attention and quantized attention under pre- and post-Hadamard multiplication scenario for Qwen2.5-3B model. The result can be found [here](https://imgur.com/a/ZzwMgOX) . We will also add it to the paper.
>
> 3. **Ablation on rank of linear correction adapters**
>
> The requested ablation is already present in our manuscript in Figure 6. We observe that Wikitext perplexity improves with larger ranks up to D = 256, beyond which gains saturate. We therefore select D = 256 as the optimal balance between accuracy and efficiency.
>
> 4. **Downstream impact of adapters on RULER**
>
> We provide downstream impact on RULER at 4K and 8K sequence length corresponding to the setting presented in Fig. 5 of our paper. Specifically, Fig. 5 isolates the impact of linear correction adapters on reducing attention error. In Table 1 below, we provide the corresponding impact of linear correction on RULER accuracy.
>
> Table 1 : RULER accuracy (higher is better) at 2-bit KV cache with and without linear correction.
>
> | Model                     | Method              | 4K       | 8K       |
> | ------------------------- | ------------------- | -------- | -------- |
> | **Llama-3.2-3B-Instruct** | 2-bit KV            | 76.7     | 70.3     |
> |                           | + Linear Correction | **77.8** | **71.1** |
> | **Qwen2.5-3b-Instruct**   | 2-bit KV            | 49.5     | 41.1     |
> |                           | + Linear Correction | **55.5** | **48.2** |
> | **Qwen3-4B-Instruct**     | 2-bit KV            | 83.7     | 79.9     |
> |                           | + Linear Correction | **84.2**     | **80.5**     |
>
> [1] Liu, Zechun, et al. "Spinquant: Llm quantization with learned rotations." _arXiv preprint arXiv:2405.16406_ (2024).
> [2] Ashkboos, Saleh, et al. "Quarot: Outlier-free 4-bit inference in rotated llms." _Advances in Neural Information Processing Systems_ 37 (2024): 100213-100240.

---

### Author Response · Authors · 2025-11-25
**Manuscript updated**

Respected reviewers,

We have updated the manuscript with requested results and clarifications. The changes are highlighted in red. Please let us know if further analysis is needed.

Thanks,

Authors of KVlinC

---

### Meta-Review · Area_Chair_Ranf · 2026-01-06

**Summary:**

The reviewers agree that the paper proposes a practically motivated KV-cache quantization method that achieves competitive performance at extreme low precision, but several core concerns limit its suitability for acceptance. First, the approach appears to offer limited novelty, as Hadamard rotation has already been explored in prior work such as QuaRot, and KVLinC primarily addresses one known weakness of QuaRot by introducing an additional trainable bias term. Second, despite being positioned alongside training-free KV-cache quantization methods, the proposed approach is not training-free, as the linear correction bias must be learned using a calibration dataset, which changes the deployment assumptions. Third, while the reported performance is competitive and often strong, the experimental evaluation is not comprehensive enough to clearly establish robustness, particularly with respect to very long-context settings, broader model scales, and more diverse baselines.

**Reviewer Concerns:**

The rebuttal addressed some clarification and ablation-related questions and added additional comparisons, but the main concerns regarding limited conceptual novelty, reliance on training rather than being fully plug-and-play, incomplete baseline coverage, and insufficiently deep long-context evaluation remain largely unresolved.

**Reviewer Scores:**

Reviewer scores would likely remain unchanged after discussion, with reviewers who were marginally positive maintaining borderline ratings and reviewers expressing concerns about novelty and evaluation continuing to lean toward rejection, supporting an overall reject recommendation.

---

### Decision · Program_Chairs · 2026-01-26

Reject